# Efficacy of Plant Sterol-Enriched Food for Primary Prevention and Treatment of Hypercholesterolemia: A Systematic Literature Review

**DOI:** 10.3390/foods11060839

**Published:** 2022-03-15

**Authors:** Elisa Turini, Miriana Sarsale, Davide Petri, Michele Totaro, Ersilia Lucenteforte, Lara Tavoschi, Angelo Baggiani

**Affiliations:** 1Department of Translational Research and New Technologies in Medicine and Surgery, University of Pisa, Via San Zeno 37, 56123 Pisa, Italy; m.sarsale@studenti.unipi.it (M.S.); michele.totaro.unipi@hotmail.com (M.T.); lara.tavoschi@unipi.it (L.T.); angelo.baggiani@unipi.it (A.B.); 2Department of Pharmacy, University of Pisa, 56123 Pisa, Italy; 3Department of Clinical and Experimental Medicine, University of Pisa, 56123 Pisa, Italy; davide.petri@unipi.it (D.P.); ersilia.lucenteforte@unipi.it (E.L.)

**Keywords:** hypercholesterolemia, phytosterols, plant sterols, enriched food

## Abstract

Plant sterols/phytosterols (PSs) are molecules with a similar structure to cholesterol that have a recognized effect on elevated LDL concentrations (LDL-c). PSs are used as a natural therapy against elevated LDL-c in combination with a healthy diet and exercise. A systematic review was performed to evaluate the efficacy of PS-enriched foods in the treatment of hypercholesterolemia. Randomized controlled clinical studies reporting the use of PS-enriched foods to reduce LDL-c among adult individuals were retrieved and assessed for risk of bias. Meta-analyses were performed to assess changes in LDL-c by treatment, food matrix, LDL-c range, sterols dosage and risk of bias (RoB). In the 13 studies analyzed, LDL-c in PS-treated participants decreased by an average of 12.14 (8.98; 15.29) mg/dL. PS administration was statistically more effective in patients with LDL-c ≥ 140 mg/dL and for PS dosages > 2 g/day. It can be concluded that PSs can be used as an important primary prevention measure for hypercholesterolemia and as tertiary prevention for cardiovascular events in patients who already have mild to moderate LDL-c. However, in severe hypercholesterolemia and in cases of familial hypercholesterolemia, it is necessary to combine dietary treatment with the use of statins.

## 1. Introduction

High plasma cholesterol, especially plasma LDL concentration (LDL-c), is associated with increased cardiovascular and cerebrovascular risk—the leading causes of death in most developed countries [1]. Therefore, control of LDL-c is an important secondary prevention measure against ischemic events. Healthcare professionals recommend a healthy lifestyle, a balanced diet and exercise to patients with raised LDL-c [2]. 

The relevance of dietary habits to the prevention of cardiovascular and cerebrovascular risk is also emphasized in the ACC/AHA Guideline on the Primary Prevention of Cardiovascular Disease. A healthy diet includes a high intake of fruit, vegetables, nuts, whole grains, lean vegetable or animal protein and fish and reduced intakes of trans fats, red meat, processed meat, refined carbohydrates and sugar-sweetened drinks. For adults with overweight and obesity, calorie restriction and weight loss are necessary. An improved diet results in lower LDL-c and improved overall lipid balance [3,4]. 

The National Cholesterol Education Program Adult Treatment Panel III recommends combined dietary therapy with a low rate of both saturated and total fat, a total cholesterol concentration lower than 200 mg/dL and consumption of soluble fiber and plant sterols/phytosterols (PS), a dose of 2 g/day of PSs being more effective. PSs (including beta-sitosterol, camp sterol and stigmasterol) have a chemical structure and cellular function similar to human cholesterol and are found in vegetable oils, nuts and seeds [5,6]. 

Several studies have consistently shown that PS-fortified foods, even as monotherapy, decrease LDL-c without any effect on HDL-c and triglycerides [7]. Therefore, PSs are often incorporated into diets through fortified foods, which can range from dairy products to bread and to spreads. PSs lower LDL-c by inhibiting both dietary and biliary cholesterol absorption. Through the intervention of bile, free cholesterol (dietary and biliary) is emulsified into micelles, lipoprotein droplets rich in phospholipids, fatty acids, bile salts and monoglycerides [8,9]. During this process, which takes place within the duodeno-digiunal lumen, PSs compete with free cholesterol for incorporation into micelles [10]. Since cholesterol can only be effectively absorbed when transported into these particles, PSs reduce absorption at this level, leading to a concomitant increase in fecal excretion of cholesterol [11]. 

Instead, at the hepatic level, PSs stimulate enzymatic adaptations to maintain cholesterol homeostasis in response to reduced absorption. First, enzymatic adaptations replace bile acid and increase hepatic cholesterol pools by increasing the expression of the rate-limiting enzyme for bile biosynthesis (cholesterol 7-a-hydroxylase) in response to reduced expression of the FXR receptor, a known suppressor of the enzyme [12]. At the same time, the rate-limiting enzyme for cholesterol biosynthesis (hepatic 3-hydroxy-3-methylglutaryl CoA reductase) is also upregulated. Second, to preserve and increase the hepatic cholesterol pool, VLDL production is reduced, as evidenced by significant decreases in plasma apoB, and hepatic LDL receptor expression increases [5]. 

If intake of PSs occurs with constancy, the cycle continues and the process of biliary and dietary absorption/reabsorption of cholesterol is blocked and fecal excretion increases. Plasma concentrations of total cholesterol and LDL continue to decline while cholesterol, which accumulates in the liver, is diverted to the bile acid pathway with which it will be excreted. The end result is an improved lipid profile: plasma concentrations of total cholesterol and LDL are decreased, and HDL and triglyceride concentrations are not altered. Therefore, there is an increase in the HDL-c/LDL-c ratio. Furthermore, consumption of PSs does not imply a high concertation of PSs in the blood because a proportion of them are excreted from enterocytes by specific transporters, and PSs that remain in the cells are transported with cholesterol to the liver by chylomicrons. PSs are then rapidly excreted through bile [5,13,14]. 

The objective of this study was to evaluate recent scientific evidence underlying the use of phytosterols for the treatment of hypercholesterolemia through a systematic review of the literature, focusing on the effect of diet on LDL-c. In particular, different dosages of phytosterols taken, different dietary matrixes and the possible different impacts of therapy according to the severity of the disease were evaluated to inform patient-centered prevention interventions.

## 2. Materials and Methods

### 2.1. Research Strategy

The research question is: “Are PS-enriched foods effective to reduce LDL-c in adult individuals affected by hypercholesterolemia?” To identify studies that examined the effects of PSs in enriched foods on plasma cholesterol, a search of the scientific literature was performed using the following key concepts: hypercholesterolemia, plant sterols and enriched foods. The search string was built using MESH terms and natural language (Appendix A, “PubMedSearchHistory 22-04”). The search was performed on PubMed, Scopus and Google Scholar during 22 April 2021. In addition, a manual search of references, including articles and relevant systematic reviews, was performed.

### 2.2. Selection Criteria

Randomized controlled clinical trials reporting the use of phytosterol-enriched foods to reduce hypercholesterolemia among adult individuals were selected. Articles in English or Spanish published from 2010 to the extraction date were included. Articles published before 2010 were excluded in order to obtain a more up-to-date review than the others and to observe whether the latest study results are in agreement with the literature. In line with previous studies, sterol dose was limited to the range of 1.5 g/day to 3 g/day [15,16]. Hypercholesterolemia was defined as LDL higher than 100 mg/dL [2]. 

Studies that did not describe the source of the PSs used, or investigated PS administration using capsules or pills, or a combination of PSs with other substances, including drugs, were excluded. Studies that did not report LDL-c at baseline and/or end of protocol were also excluded. The laboratory method for the measurement of cholesterol was not considered relevant for selection purposes, as long as it was unchanged during the conduction of each specific study. Finally, participants’ dietary habits were considered irrelevant for study selection, as we assumed that they did not affect the baseline values.

The three reviewers (ET, AM, MS) piloted the selection criteria on a subset of articles obtained from the title/abstract screening. After reaching 99% concordance, they proceeded with single-rater selection in full text for the remaining studies. For the full text selection, studies that were published before 2010 reporting on foods enriched with more than one therapeutic molecule in addition to PSs that did not specify the LDL-c range and did not describe the daily dose of sterols or had a range outside our criteria were excluded.

For the design and development of the manuscript, the PRISMA guidelines were followed and the PRISMA checklist was compiled (see Appendix A, “PRISMA_2020_checklist). The study was registered on the International Prospective Register of Systematic Reviews, PROSPERO, on 06/08/2121, ID 272061, CRD42021272061.

### 2.3. Data extraction and Quality Assessment

Characteristics of the included studies were extracted and collected in a spreadsheet. The set of variables was: year of publication and study period (if indicated), name of first author, type of study, range or mean age of participants, gender, matrix, daily dose of PS, type of PS, baseline and post-intervention values, first for PS-treated groups then for placebo/control groups. If a study reported LDL ranges with units in mmol/L, a conversion from mmol/L to mg/dL was performed. If a study showed only the difference between the baseline and end values with its confidence interval, we obtained the end value by subtracting the difference from the baseline value, whereas for the standard deviation we obtained standard deviation of the difference (*SDdiff*) from the proposed 95% confidence interval; then *SD*2 was derived using the following formula: *SDdif f*^2^ = *SD*1^2^ + *SD*2^2^ − 2 × *corr* × *SD*1 × *SD*2(1)
where the correlation was assigned a value of 0.8. Studies reporting ranges of LDL values, including SEM values, were adjusted to SD using the formula SEM = SD/√n. LDL data reported as medians with minimum and maximum values both baseline and post-FU were converted into means and variances using the formulas reported by Hozo et al. [17].

The methodological quality of each selected study was assessed according to the RoB 2.0 tools for parallel and crossover studies [18] depending on the study design. The tool considers five domains (randomization process, deviation from intended interventions, missing outcome data, measurement of the outcome and selection of the reported result), each rated in terms of risk of bias and applicability to the research question. RoB was performed in a design-specific way, assigning each study to the appropriate tool. Risk of bias (RoB) was judged as “low”, “some concerns” or “high”. Each domain included different signalling questions guiding the risk of bias assessment. If all signalling questions received a favourable answer, then the risk of bias was judged as “low”. At any review stage, disagreements were resolved by discussion or by the involvement of a third investigator (data not shown; see Appendix A, “ROB2_crossover_beta_v1” and “ROB2_IRPG_beta_v8-Foglio buono”).

### 2.4. Data Analysis

The data for the included studies were entered in Review Manager Software Review Manager (RevMan) Version 5.4, the Nordic Cochrane Centre, The Cochrane Collaboration, Copenhagen, Denmark, 2020. The meta-analyses were performed with the creation of forest plots. For the analyses, the mean difference (MD) of individual studies between LDL-c measured at the end of follow-up versus LDL-c measured at baseline was calculated. The values of the individual studies thus derived were subsequently analyzed as a whole using the inverse variance method to calculate weights, and the statistical model adopted was a random-effects meta-analysis.

Statistical heterogeneity between studies and groups was assessed applying Cochran’s Q-test. An I^2^ statistic was reported as a quantification of a study’s heterogeneity. A *p*-value < 0.10 was considered as indicative of statistical heterogeneity.

The main meta-analysis was performed to compare LDL-c between treated and placebo groups. A set of sub-analyses or stratifications were conducted to compare: (i) type of food (liquid/solid); (ii) LDL-c range at baseline using 140 mg/dL as a cut-off value (this cut-off was pragmatically defined based on the range of extracted LDL values); (iii) sterol dosage; and (iv) RoB assessment.

## 3. Results

### 3.1. Studies Description

A total of 362 articles were identified from the search strategy on PubMed, and 107 articles from Scopus, Google Scholar and other sources. An initial selection was made by examining the title and abstract, resulting in 269 studies. After full-text selection, a total of 13 studies were included for further analysis (Figure 1).

The included studies were carried out in Australia [19], Brazil [20], China [21,22], Finland [23,24], Germany [25], Indonesia [26], the Netherlands [27], Spain [28,29,30] and Turkey [31], with the large majority (8, 61.5%) in Europe. All the studies were written in English, except one [30]. The included studies were published between 2010 and 2020, and all studies were conducted after a randomization process to form two groups, treatment and placebo, and had a follow-up period of 3 weeks to 6 months depending on the study. Only one study [20] was single-blinded, while the others were all double-blinded. According to the risk of bias evaluation, all studies were found to be low risk, except for three with some concerns [23,24,29] and one high risk [20] (Appendix A; see Appendix A).

The participants were adult with an overall average age of 44 years (range 18–70). The study population had heterogeneous baseline cholesterol values, ranging from mild hypercholesterolemia, with a minimum value of 116.6 ± 22.78 mg/dL among all baseline values, to high hypercholesterolemia, with a maximum value of 178.76 ± 31.35 mg/dL among all baseline values. In one study [28], participants in the control group were administered omega-3 (Appendix A).

The daily dose of food enriched with PSs was not reported in all studies [19,24]. The minimum sterol dose administered was 1.57 g/day and the maximum 3 g/day. The matrixes were dairy products, such as yoghurt, milk and milk powder, soy powder, soy milk and margarine, bread, spreads and smoothies. The PSs were all of plant origin and were used in esterified form in five studies [22,24,26,27,31] and in non-esterified forms or in the form of phytosterols in the remaining studies. In three studies [25,28,29], the types of sterols used were specified: sitosterol, campesterol and stigmasterol (Appendix A.

### 3.2. Impact of Foods Enriched with PSs

As shown in Figure 2, LDL-c in the PS-treated groups decreased in all studies, with an average of 12.14 mg/dL (8.98; 15.29) less than the baseline value (Figure 2). 

No variations in LDL-c were observed among participants in the placebo/control groups (Figure 3).

A sensitivity analysis was performed, stratifying by RoB categories. The analysis shows that there were no differences in outcomes between the studies with low risk of bias and those with medium/high risk. Therefore, the latter were not excluded (data not shown; see Appendix A, “Forest plot”).

### 3.3. Effect of PS Dosage

Daily dosage of PSs was evaluated to assess whether it effected a decrease in LDL-c. A dose of more than 2 g per day was more effective (*p* = 0.01) than lower or equal doses (Figure 4). 

Finally, in order to evaluate the type of patient for whom the use of PSs was most indicated and most effective, the administration of PSs in participants with LDL-c ≥ 140 mg/dL and LDL-c < 140 mg/dL was investigated. The treatment with fortified foods was effective independently of baseline values, but the decrease was significantly higher (0.01) among the group with LDL-c ≥ 140 mg/dL (Figure 5).

### 3.4. Impact of Matrix

According to the sub-analysis of matrix status, the use of solid enriched foods was associated with more heterogeneity as compared to studies reporting on liquids. There was a marginal difference in efficacy in favor of the latter, although not statistically significant (0.64). (Figure 6).

## 4. Discussion

This systematic review evaluated the impact of a diet enriched with PSs on the treatment of hypercholesterolemia by assessing changes in participants’ LDL-c. PSs taken via fortified foods were found to be effective in significantly reducing hypercholesterolemia, in line with previous reviews on the subject [5,6]. Some variability in LDL-c reduction was observed among studies. This may be explained by various factors, such as differences in dietary fatty acid composition, source of PSs, timing of PS ingestion, basal LDL-c and genetic differences between individuals [10,32]. Underlying dietary regimens may also have contributed to heterogeneity, as in a few studies participants in treatment/control groups were advised to follow a healthy diet instead of their usual one [5]. However, the meta-analysis showed that, independently of diet regimen, no significant changes in LDL-c were observed in the placebo group. Only in one study a significant drop in LDL-c was reported among the placebo group [25]. However, no information was provided with regard to participants’ dietary regimens. On the contrary, little variability could be attributed to the type of food matrix. As shown by the analyses and confirmed by previous studies, the solid or liquid composition of the matrix did not significantly influence LDL-c. Existing evidence suggests that matrix does not affect intestinal sterol absorption [10]. Likewise, gastric absorption of sterols, as with cholesterol, is irrelevant [10,33]. The success of PSs is attributable to inhibition of intestinal absorption of both dietary and biliary cholesterol and increased fecal excretion of cholesterol, which promotes the utilization of endogenous cholesterol [33]. PS intake has been shown to be safe [34] and without side effects or changes in the absorption of other substances [35]. There are no changes in HDL-c and triglycerides caused by PSs [36,37]. This was confirmed by our study, as no side severe effects were reported, only diarrhea, flatulence and dyspepsia in two studies [21,26]. According to the present analysis, the effect of PSs was influenced by baseline LDL-c. PS-induced LDL-c reduction was significantly higher among patients with LDL-c ≥140. Participants in the studies included in this review suffered from secondary hypercholesterolemia and it can be assumed that elevated LDL-c is caused more by an unhealthy, high-fat diet. As previous studies have suggested, the efficacy of PSs in reducing blood LDL-c is greatest where there is an excessive dietary intake of fats [8]. This is the reason why the subjects involved in this study benefited more from PSs when LDL-c was elevated. On the contrary, if the diet contains low amounts of fat but cholesterolemia remains high, being mainly caused by genetic factors, the effectiveness of sterols is limited. In fact, hypercholesterolemia may persist if the body produces excess cholesterol, even with exercise, the correct diet and PS intake [38]. When patients suffer from familial hypercholesterolemia, dietary measures have a restricted potential and the use of statins is necessary [39]. In addition, individuals with elevated LDL-c also have higher levels of serum cholestanol, a marker of cholesterol absorption. Patients who respond best to phytosterols are those with higher levels of cholestanol [40]. Cholestanol concentration has not been researched in the patients of the pooled studies and is a potential marker to be investigated in order to evaluate the subgroups to be offered PSs with greater effect. These statements further underline the importance of the intestinal absorption mechanism of both exogenous hypercholesterolemia and the efficacy of PSs, which work at this level by significantly decreasing it [41]. Finally, the present findings are in line with the NCEP/ATP III recommendation that administration of PSs < 2 g/day is less effective in reducing LDL-c as compared with higher doses. A significant cholesterol-lowering effect requires at least 2 g/day, although the causes remain to be clarified. Previous studies suggest that the efficacy of phytosterols is dose-dependent. In this study, a plateau effect was observed above a daily dose of 3 g/day [42], suggesting that no positive effect in reducing cholesterol could be achieved with higher dosages [16]. This could be due to the fact that a high intake of PSs does not improve the systemic availability of plant stanols [43]. However, a previous review demonstrated reported clinical evidence of a positive effect until 10 g/day [44]. Beyond PSs, there could be other non-synthetic alternatives for the treatment of hypercholesterolemia. Cicero et al. [45] compared the efficacy of PSs with that of red rice, which is a molecule known to have cholesterol-lowering effects, rather than with placebo. Cicero’s study showed that red rice is much more effective than PS, highlighting that PSs are not the only non-pharmacological alternative for lowering LDL. Selection bias could be a limitation of this study: all possible studies could not be identified, the included studies did not report all information in detail and sample size and composition were not necessarily comparable, nor were dietary regimens taken into account. Finally, one of the included studies [29] did not use a placebo but omega-3s instead. These have no proven action against elevated LDL-c, so omega-3 may be considered a placebo for the purposes of this review. The strengths of this work are the lack of heterogeneity and results that are in line with the literature on the topic.

## 5. Conclusions

It can be concluded that the use of PSs in combination with a healthy lifestyle in patients with mild to moderate hypercholesterolaemia reduce LDL-c. In these patients, PSs can be used as primary prevention of cardiovascular and cerebrovascular events. However, in severe hypercholesterolemia and in cases of familial hypercholesterolemia, statins must be used in addition to dietary treatment, as use of PSs is insufficient. 

## Figures and Tables

**Figure 1 foods-11-00839-f001:**
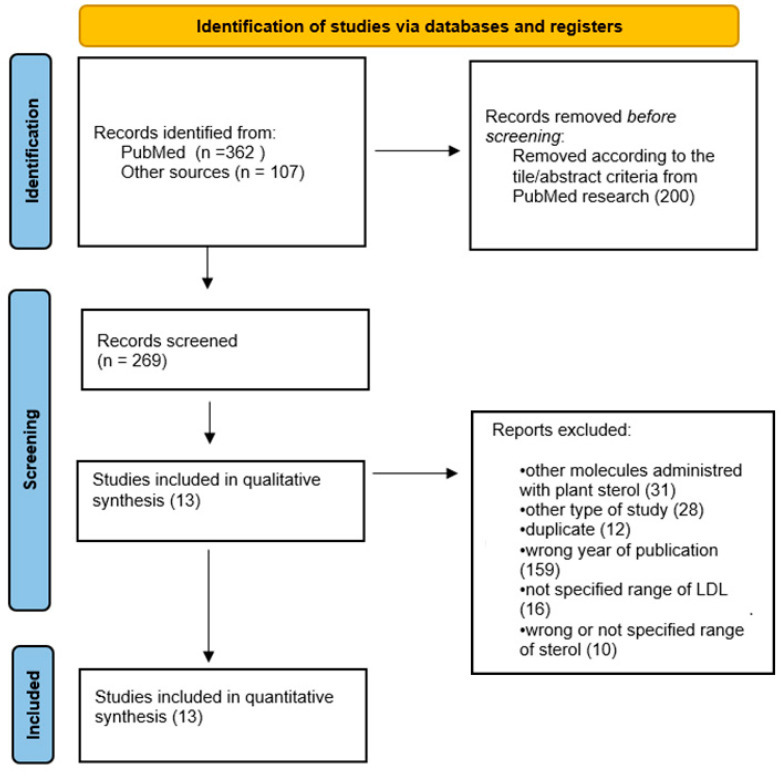
PRISMA flow chart of included studies.

**Figure 2 foods-11-00839-f002:**
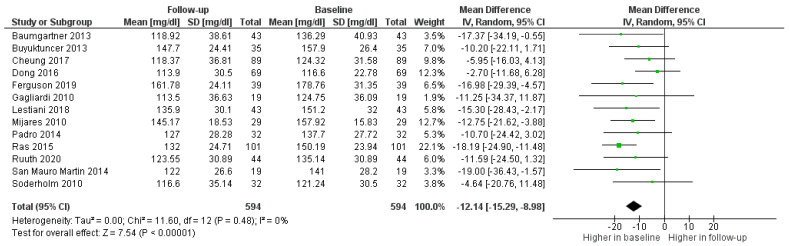
Forest plot comparing baseline and follow up LDL values among treated participants.

**Figure 3 foods-11-00839-f003:**
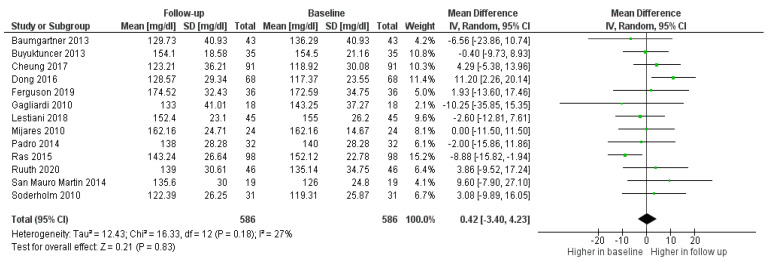
Forest plot comparing baseline and follow up LDL-c among participants in the control/placebo groups.

**Figure 4 foods-11-00839-f004:**
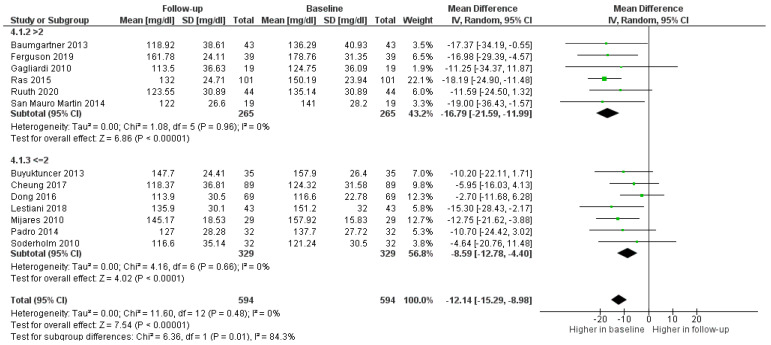
Forest plot comparing different dosages of PSs among participants in the treatment groups.

**Figure 5 foods-11-00839-f005:**
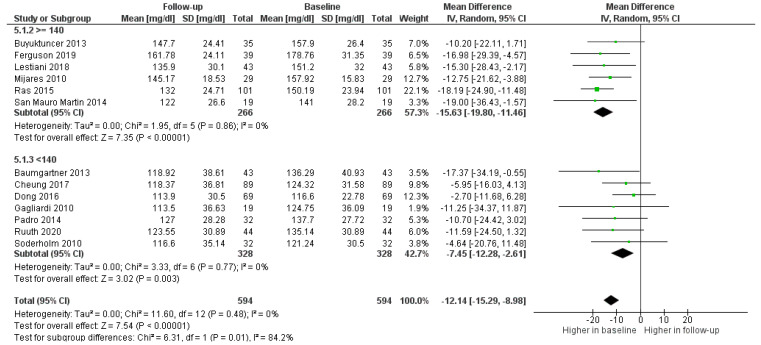
Forest plot comparing different ranges of LDL-c among participants in the treatment groups.

**Figure 6 foods-11-00839-f006:**
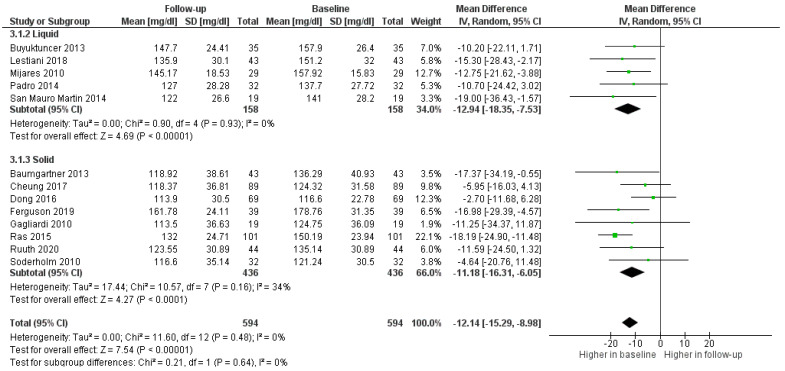
Forest plot comparing liquid or solid matrixes among participants in the treatment groups.

## Data Availability

Not applicable.

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
