# Peer review of "Efficacy of Plant Sterol-Enriched Food for Primary Prevention and Treatment of Hypercholesterolemia: A Systematic Literature Review"

_foods, 2022, doi:10.3390/foods11060839_

Round 1

Reviewer 1 Report

In the introduction as well as in the discussion, the importance of dietary habits in modifying cholesterol levels should be further discussed.

Justification should be given for including studies more than 10 years old in the review.

The conclusions of the study are not correctly formulated.

Reviewer 2 Report

The authors conducted a review that was to evaluatethe efficacy of plant sterol enriched foods in the treatment of hypercholesterolemia, including different dosages of phytosterols taken, different dietary matrix and the possible different impact of therapy according to the severity of the disease. The review was carefully prepared and the presented studies are interesting and focused on the main topic.
This work is really interesting and represents a meta-analysis of the  literature on this topic and I believe it is delegante for this field of research and for the potential translability of the collected data into the clinical practice.
But as in this type of work some mistakes have been made: 
1. Not all associeated factors are shown that may be relevant in  primary preventions and treatment for hypercholesterolemia. This is worth mentioning in the limitations of the study.
2. Maybe it is worth adding more keywords, which will allow for better recognition and citation of the article. 
3. In the objective of this study (line 78) should be specified what kind of diet is it - with PS? 
4. Figure 1 should be aligned with PRISMA guidelines, 
as well as the titles and captions of the figures - should be  under the figures.
5. References should be adapted to the recommendations (journal guidelines). In addition, Authors should check cross-references in the text, e.g. line 134 - Hozo et al is not item 15, similarly item 16 - it should be 19 [Sterne et al] etc.  
However, despite the above-metioned accusations I rate the work higly but Authors must read it carefully and correct. 
